# Dependence of the Nanoscale Composite Morphology of Fe_3_O_4_ Nanoparticle-Infused Lysozyme Amyloid Fibrils on Timing of Infusion: A Combined SAXS and AFM Study

**DOI:** 10.3390/molecules26164864

**Published:** 2021-08-11

**Authors:** Martin A. Schroer, Po-Sheng Hu, Natalia Tomasovicova, Marianna Batkova, Katarina Zakutanska, Po-Yi Wu, Peter Kopcansky

**Affiliations:** 1European Molecular Biology Laboratory, Hamburg Outstation c/o DESY, Notkestr. 85, 22607 Hamburg, Germany; 2College of Photonics, National Yang Ming Chiao Tung University, Tainan City 71150, Taiwan; power528520@gmail.com; 3College of Photonics, National Chiao Tung University, Tainan City 71150, Taiwan; 4Institute of Experimental Physics, Slovak Academy of Sciences, Watsonova 47, 04001 Kosice, Slovakia; nhudak@saske.sk (N.T.); batkova@saske.sk (M.B.); zakutanska@saske.sk (K.Z.); kopcan@saske.sk (P.K.)

**Keywords:** nanocomposites, biological–nanoparticle hybrid systems, lysozyme amyloid fibrils, small angle X-ray scattering (SAXS), atomic force microscopy (AFM)

## Abstract

Understanding the formation process and the spatial distribution of nanoparticle (NP) clusters on amyloid fibrils is an essential step for the development of NP-based methods to inhibit aggregation of amyloidal proteins or reverse the assembling trend of the proto-fibrillary complexes that prompts pathogenesis of neuro degeneration. For this, a detailed structural determination of the diverse hybrid assemblies that are forming is needed, which can be achieved by advanced X-ray scattering techniques. Using a combined solution small angle X-ray scattering (SAXS) and atomic force microscopy (AFM) approach, this study investigates the intrinsic trends of the interaction between lysozyme amyloid fibrils (LAFs) and Fe_3_O_4_ NPs before and after fibrillization at nanometer resolution. AFM images reveal that the number of NP clusters interacting with the lysozyme fibers does not increase significantly with NP volume concentration, suggesting a saturation in NP aggregation on the fibrillary surface. The data indicate that the number of non-adsorbed Fe_3_O_4_ NPs is highly dependent on the timing of NP infusion within the synthesis process. SAXS data yield access to the spatial distribution, aggregation manner and density of NP clusters on the fibrillary surfaces. Employing modern data analysis approaches, the shape and internal structural morphology of the so formed nanocomposites are revealed. The combined experimental approach suggests that while Fe_3_O_4_ NPs infusion does not prevent the fibril-formation, the variation of NP concentration and size at different stages of the fibrillization process can impose a pronounced impact on the superficial and internal structural morphologies of these nanocomposites. These findings may be applicable in devising advanced therapeutic treatments for neurodegenerative diseases and designing novel bio-inorganic magnetic devices. Our results further demonstrate that modern X-ray methods give access to the structure of—and insight into the formation process of—biological–inorganic hybrid structures in solution.

## 1. Introduction

The progressive deposition of amyloidal fibrils over a long period in one’s life time was postulated as a major cause of some chronic life-threatening diseases such as type II diabetes [1], atherosclerosis [2], or neurodegenerative disorders [3]. These detrimental extracellular fibrils, made of assemblies of protofibrils in the formation of 4.8 angstrom misfolded or partially misfolded polypeptides through interaction among fibrillary surface-bound hydrogen bonds, were characterized by reverse cross-β sheet orthogonal to the axis of fibrillary length [4]. With over 100 types of diseases found to be associated with wrongly folded proteins of various origins, these oligomers—or fibrillary deposition, as a whole, can erupt cells’ lipid membranes, deregulate the process of genetic transcription, de-functionalize mitochondria, and disrupt synaptic plasticity, among other effects [5]. Hence, the need for a cure for the prevention, reversal of pathogenesis and extermination of the toxic aggregating monomeric elements or fibrillary complexes has opened up a brand-new avenue of research direction, and spurred a plethora of research studies that counteract the diseases by devising better preventive measures, and more effective pharmaceutic agents and therapeutic methods [6].

Nanomedicine that comprises nanomaterials such as iron oxides (Fe_3_O_4_), gold (Au) and zinc oxide (ZnO) nanoparticles (NP), to name just a few, received significant attention from the research community due to its unique biochemical and physical properties, and was verified to be instrumental in facilitating various magnitudes and manners of interaction with amyloidal β (Aβ) fibrils [7,8,9]. Atop random interactions, many aspects of NPs, including their sizes, shapes, surface charges, pH-dependence, volume concentration, hydrophobicity and surface modification, were exploited for their potential in disintegrating or reversing the trend of fibrillary aggregation [10,11,12,13,14,15,16].

Multi-functional characteristics of Fe_3_O_4_ NPs, including magnetic, photothermal, and catalytic properties, are especially popular as biomedical reagents in the treatment of fibrillary aggregation-induced phenomena. The advantage of iron oxide NPs is their low toxicity when compared with other metal oxide NPs. The demonstrated inhibition and destruction of the amyloidal complexes of many protein origins, such as lysozyme and insulin, are highly depend on many factors, including, but not limited to, concentration and size, or surface treatment [7,8,9,10,14]. However, the use of iron oxide NPs in amyloid protein aggregation is still limited.

Up until now, several studies have proven the effectiveness of the Fe_3_O_4_ NPs, with or without surface modification, in imposing a destructive or agglomerating influence upon Aβ fibrils [7,8,9,10,17,18,19,20,21,22,23]. It was shown that, in biological media, nanoparticles interact with different proteins. However, several aspects are still not fully understood, for instance, the dependence of the internal molecular morphology, aggregating manner and spatial distribution of NP clusters at the fibrillary surface on the timing of infusion. In particular, the morphological quantification of such interaction within the nanocomposites is relatively unknown.

The process of nanoparticles’ adsorption on the protein surface can be influenced by several factors. More recently, in our previous work [23], the structural morphology of nanocomposites made from lysozyme amyloid fibrils (LAFs) and Fe_3_O_4_ NPs was studied to find the optimal conditions to prepare stable magnetic-responsive nanocomposite materials composed of fibrils and nanoparticles, which might allow manipulation by external magnetic fields. It was shown that the stability depends on the ratio of LAFs and Fe_3_O_4_ NPs. This study further demonstrated that solution small angle X-ray scattering (SAXS), in combination with further experimental methods, allows the study of the structural morphology of such nanocomposites.

Following our previously established approach, the aim of the present work was to elucidate the influence of nanoparticle size and concentration on the formation of fibril-nanoparticles complexes and its dependence on the infusion time. Using nanoparticles of different sizes and concentrations, the distribution patterns, clustering manner and interactive morphology of the resulting fibrillary nanocomplexes were studied by solution SAXS and AFM measurements. While the SAXS data allowed the characterization of the shape and internal structural morphology of the nanocomposites and yielded structural models of these, AFM images were taken to ensure mutual interactions between LAFs and Fe_3_O_4_ NPs, and thus to elucidate the manner and morphology of the aggregates of NP clusters on the fibrillary surface. The understanding of such NP–protein interactions is the first step towards deciphering the true nature of the NP-mediated biological and biomedical effects and might lead to strategic manipulation in vivo. Our study demonstrates how modern solution SAXS approaches can reveal such structural information on biomaterials.

## 2. Results

### 2.1. AFM Imaging

Mutual interaction between the NPs and LAFs, and thus, the manner of NP adsorption on the fibrillary surface of the pure and composite fibrillary samples were determined by AFM imaging. AFM images of pure LAFs at different magnifications are shown in Figure 1. The average diameter is 10 nm, while the length varies between several hundred nm to 2 µm.

To study the dependence of the fibrillary interaction with 10 nm NP on volume concentration, 10 μL and 30 μL of the NP solution were infused separately with 300 μL of LAFs, and acquired with AFM imaging as shown in Figure 2. The scale bars are (a) 4 µm, (b) 2 µm and (c) 500 nm. 

The figures at the highest resolution show NPs adsorbed on the LAFs, confirming the binding affinity between the fibrils and 10 nm NPs. These results are in accord with previous studies, which reported such an attachment of NPs on the fibril surface [21,24,25,26]. For the higher volume of NP added (30 μL), there appears to be more unattached NPs than for 10 μL.

When 10 nm NPs were added to the solution before the fibrillization process was started (type-B process), LAF formation was observed as well as NP adsorption. The morphology of the aggregated NP clusters on the fibrillary surface, shown in Figure 3, exhibits larger and thicker agglomeration than those formed in the type-A process. As the volume concentration increases, freely floating NPs are also proportionally increasing, implying a saturation in the feature size of the nanocomposites despite an increase in the volume concentration. To explore the role of NP size on the fibrillary interaction, measurements using 20 nm NPs were also carried out. Figure 4 reflects a clear dependence of the random NP attachment with the pre-formed LAFs. While the sites of NP clusters on the fibrillary surfaces indeed slightly increase, the feature size of NP clusters of 300 μL is nominally comparative to that of other volume concentrations, confirming the result of 10 nm NPs, namely that a large volume concentration does not significantly change the cluster size and morphology.

By carrying out the fibrillary interaction with 20 nm NPs before the fibrillization (type-B process) (Figure 5), one can find a marked augmentation in the numbers of freely unattached NPs and the attachment sites as the volume concentration increases from 30 μL to 300 μL in comparison to the type-A process, while no striking increase in the size of NP clusters can be identified.

Subsequently, the size of the NPs was increased to 30 nm. Figure 6 and Figure 7 present the fibrillary nanocomposites made of the LAFs and 30 nm NPs by the type-A and type-B processes, respectively. Since 30 nm NPs have a larger size and correspond to a smaller number of particles in an equivalent volume, both types of processes allow fewer binding sites of NP clusters on the fibrillary surface than those of 10 nm NPs and 20 nm NPs. As a consequence, null unattached NPs can be found in the images. One stark contrast between the two processes is the larger thickness of the nanocomposites associated with the type-B process, and the manner of coverage of NP clusters upon the fibrillary surface is morphed into either lamellar conformation or point-like attachment. Generally, although the corresponding measured values of the zeta potential of pure LAFs (44.8 mV), and of Fe_3_O_4_ NPs of 10 nm, 20 nm and 30 nm (−55.4 mV, −31.5 mV, and −37.1 mV), are opposite and similar in the range of magnitude, the numbers of un-adsorbed NP clusters are proportionally higher as the concentration increases, implying a limiting saturation on the number of NP clusters on the fibrillary surface. In addition, a noticeable difference is the association of less floating NPs with the type-A process as opposed to the type-B process.

### 2.2. SAXS Characterization

Following the 2D morphological characterization of the nanocomposites by AFM imaging, SAXS data were collected for the same sets of freshly prepared samples in solution. In the first step, the SAXS profiles of the pure species, LAF and NPs of various sizes, were studied and are shown in Figure 8a. The SAXS curve of LAFs is typical for the form factor of an elongated particle, exhibiting a power-law decay at smaller angles, and shows some characteristic modulation at *q* = 0.9 nm^−1^. In addition, the corresponding pair–distance distribution function *p*(*r*) of the pure LAFs (presented in Figure 8b) was determined using the GNOM program [27] of the ATSAS software package [28]. The maximal particle size of the averaged LAF is *D_max_* = (80 ± 2) nm; however, due to the limited resolution of the SAXS instrument, the larger species, which are likely present in solution, cannot be resolved properly.

The profile of the *p*(*r*) function reflects the elongated shape of the fibrils with an effective cross-sectional diameter of *d* = (23 ± 2) nm. From the SAXS data, an ab initio shape model of the average solution structure of LAF was determined using the ATSAS program DAMMIF (inset, Figure 8b) [29], which is represented by an assembly of dummy beads (blue). This model exhibits a bent and partially twisted appearance and is similar to the finding for the LAFs prepared by the same method and with the same physical conditions in our previous study [23]. From the pronounced oscillations of SAXS curves for the NPs, the spherical shape and a rather low dispersity were manifested. The shift of the sets of minima to smaller *q*-values consistently revealed the increase in particle size from P10 to P30. From the SAXS data, the volume distribution function *D_Vol_*(*r*) for polydisperse spheres was computed with GNOM (Figure 8c), from which the average radii were determined as P10: 〈R〉 = (5.7 ± 0.1) nm; P20: 〈R〉 = (10.7 ± 0.1) nm; P30: 〈R〉 = (13.3 ± 0.4) nm. These values are reasonably consistent with the nominal sizes provided by the manufacturers. A summary of the extracted structural parameters for the pure material species is tabulated in Table 1.

With the species of pure materials being characterized, the fibrillary nanocomposites made of LAFs and the NPs were examined to investigate the morphological effects of varying sizes and volume concentrations of NPs and types of mixing processes. Figure 9a depicts the SAXS curves for the mixtures with P20 for both types of mixing in the *q*-range from 0.05 nm^−1^ to 1.0 nm^−1^. In general, with increasing of the NP concentration, the contribution of the P20 form factor significantly contributes to the scattering signal, most prominently seen by the presence of the characteristic oscillations. The significant signal of P20 at a rather low mixing ratio of 10/300 (NP/LAF) for samples B10 and A10 stems from the much stronger scattering power of Fe_3_O_4_ compared to the weaker scattering by lysozyme protein (~100 times less), which contributes at smaller scattering angles [23]. A direct comparison between both sets of samples reveals the most significant discrepancy at small scattering angles (*q* < 0.2 nm^−1^), while the rest of the curves look similar. In order to analyze how significantly the scattering curves from the simple mixture of both species in solution differed, all profiles were fitted as a linear combination of the pure LAF and P20 data using the ATSAS program OLIGOMER [30] (dashed lines).

For the samples made through the type-A process, the model of an oligomeric mixture fits the two lowest concentrations (A10, A30) well, while for A300, there are slight deviations present for *q* < 0.15 nm^−1^. In particular, the experimental data exhibit a weak but visible shoulder in this angular range, indicating the presence of an additional structural factor contribution. Such an interference signal might imply the formation of clusters or the interactions between the fibrils and the nanoparticles [31,32,33,34,35].

Likewise, for the samples made through the type-B process, deviations from the simple mixture model are present for all concentrations. For the low concentrations (B10, B30), the curves exhibit a steeper decay, indicating the formation of larger species. The curve for sample B300 exhibits both a stronger scattering intensity at small *q* as well as the presence of the contribution of a rather broad structure factor.

For the mixtures of LAF with P10 and P30 (Figure 9b), only the lower concentrations were studied. For P10, a similar scenario is present as for P20, with the type A following a mixture model, while the type-B deviates at small *q*-values. For P30, small deviations are present for both series, but to a lesser extent. While the modelling indicates the presence of additional structures in solution, it also shows that LAFs are formed for both processes. Thus, consistent with the AFM results, the time of infusion does not inhibit the fibrillization.

In order to determine how significantly the experimental SAXS data deviated from a simple superposition of pure LAF and NPs, an alternative analysis was performed to verify how many independent components could be extracted from the sets of SAXS curves shown in Figure 9 using the ATSAS program SVDPLOT, a method of singular value decomposition [30]. The SVD analysis of the different concentration series yields about three components, which implies the presence of a third species in solution besides the pure LAF and NPs. Note that these species are different for the varying conditions. In order to identify the scattering signal for such structures, which might explain the deviations of the SAXS curves, particularly at small angles, and following our previous approach [23], the respective datasets were analyzed using the ATSAS program DAMMIX [36]. This approach allows one to restore the shape of an unknown component in an evolving system together with the volume fractions. Here, it was assumed that the initial and final states were composed of pure NP and LAF solutions, respectively, and that the evolving parameter was the NP/LAF fraction, and thus the average number of LAF with incorporated NPs as the evolving species.

Figure 10 displays the experimental SAXS curves with the fits of the DAMMIX analysis. With the additional evolving component for each dataset, the improved fits can be obtained. In particular, the curve can now be well described at smaller *q*-values.

The computed scattering curves for all samples from the so determined evolving species of the nanocomposites are displayed in Figure 11a. For all samples of the type-B process and P20 of type-A process, the SAXS profiles exhibit the characteristic minima of the spherical form factor plus a power-law decay in the range of small angles with some additional weak modulations. As such, these curves contain the essential features present for a partial linear arrangement of clustered nanoparticles [34]. In contrast, the curve determined for P20A is similar to a core-shell object of ~40 nm radius. For both types of structures, modulations in the range of lower *q* likely contribute to the observed weak structure factor-like contribution in the experimental data. Note that for the A-series of P10, the DAMMIX analysis yielded a third component with zero weight, consistent with the previous finding of the OLIGOMER analysis that the data are well described as a simple superposition of LAF and P10 without any significant intermediates formed. The structural differences are more obvious when comparing the corresponding *p*(*r*) functions. In this real-space representation, one can clearly distinguish the elongated character for all type-B samples (solid lines). These appear as the formation of elongated assemblies of the respective NPs with an overall length of 80 nm, which corresponds to the *D_max_* of LAF. Given the stronger scattering strength of the NPs than the fibril, the data suggest that the evolving species for all the type-B-processes are NP adsorbed or clustered on the fibril axis. For the type-A process, the findings are more heterogeneous. For P10, no intermediate is formed, while for P30 the pair–distance distribution function is very similar to the type-B sample. For P20, the *p*(*r*) resembles that of a hollow-sphere with diameter of 70 nm formed by the NPs, implying that it is wrapped around the fibrils, which again, because of the weaker scattering contrast, can be difficult to observe.

### 2.3. Ab Initio Models

Based on the curves for the intermediate clusters shown in Figure 11a,b initio shape models were computed using the ATSAS program DAMMIN (Figure 12) [37]. All models for type-B process resemble a partial linear assembly of nanoparticles with clearly distinguished radii. These models are well aligned with the LAF model, supporting the assumption that the clusters are formed on these fibrils. The average number of NPs within these clusters varies from ~10 (P10) to ~5 (P20) to ~2 (P30), which scales with the NPs’ size and indicates, on average, a linear relation between the composition of clusters and the particle radius.

The model for the intermediate state of P20A appears as a spherical assembly of ~8 NPs, which, when aligned with the LAF model, suggest adsorption on it from several sides. The model for P30A (not shown) is similar to that for P30B, reflecting their similarity in the scattering curves.

Finally, the volume fractions of the three species, LAF, NPs, and intermediate composite as a function of NP/LAF ratio as determined from the DAMMIX analysis are presented in Figure 13. The fractions of LAFs and NPs decrease and increase, correspondingly, when the ratio increases, while the volume fractions of the intermediate clusters are relatively stable, within 5–15%, indicating an overall weak concentration dependence of the cluster formation. Moreover, for the type-B process, the volume fractions of the intermediates are slightly higher than those of the type-A process at low to moderate mixing fractions. For the highest mixing ratio (P10 data), the type-A process yields a larger concentration of intermediates.

## 3. Discussion

The presented study investigated the surface topography and internal structure of the nanocomposites formed by LAF and Fe_3_O_4_ NPs of different sizes as well as the role of the mixing time point of NPs during the fibrillization process. It revealed how the formation process can influence the spatial distribution of NP clusters adsorbed on the fibrillary surface and the composition of the final internal structures of the composites. AFM images were acquired to examine the mutual interaction between the NPs and LAFs as well as the aggregating manners of NP clusters on the fibrillary surface and topological patterns of the nanocomposites. Using SAXS measurements and modern data analysis schemes, the presence of nanocomposites in solution were confirmed and their actual solution structures were revealed.

Combining these experimental techniques, specific structural differences in the formation processes of the nanocomposites were determined. Unlike the type-B process, in which the number of un-adsorbed NPs decreased as the particle size increased, very few to null un-adsorbed NPs could be found in the cases of the type-A process. As manifested in the AFM images for NPs of 10 nm, thicker composites were formed than for larger NPs (20 nm and 30 nm). Generally, for both process types, a larger volume concentration of NPs does promote slightly more NP clusters; however, the freely unattached NPs also proportionally increase, suggesting a limited saturation for the integration between the NPs and LAFs. The two significant features of the type-B composites are the higher number of unadsorbed NPs and the larger cluster size than those of type-A composites, which are present for all three NP sizes. Moreover, the type-A process generally produces more NP clusters as the concentration increases, which is not pronounced in the composites by the type-B process. The clusters of 30 nm NPs attached to the LAF appear less abundant and are not significantly larger than those of other NP sizes, which is reasonable due to the fact that the numbers of 10 nm NPs and 20 nm NPs are at least 60 times that of 30 nm NPs.

As an alternative to the topographical morphology of the nanocomposites measured by AFM, SAXS measurements allowed the determination of their formation and structure in solution. Based on the detailed analysis, the presence of a fraction of nanocomposites in solution besides pure LAFs and NPs was confirmed. Moreover, ab initio modeling determined the overall average structure of these nanocomposites and their volume fraction as a function of infusion time, mixing ratio and particle size. The resulting models provide insight into the characteristic features of the formed nanocomposites. One noticeable feature of type-A and -B samples is the distinguishable presence of NPs of the respective sizes, and the well-aligned composite structures between the two models in pure and intermediate states. The final morphology of NP agglomeration for the type-B process appears patchy and partially continuous on the entire fibrillary surface, whereas in the type-A process, randomly generated clusters can form on the surface with less adherence to the fibrillary body. This is an assembling morphology that cannot be observed in the topographical presentation of AFM images. In addition, the average number of NPs within these agglomerated clusters increases as the particle size decreases, suggesting a linear relationship between the area of surface coverage and radius of NPs, which was also not clearly elucidated from the AFM analysis on a small portion of the composite solution with a finite number of objects deposited as a thin film.

Finally, the volume fractions of pure NPs, pure LAFs and intermediate states were obtained from the modelling. Although the SAXS results did not convey a strong indication that the number and cross-sectional width of NPs clusters are highly dependent on the volume concentration, as suggested by the AFM analysis, such precise estimation of the number of NPs binding on the fibrillary surface is not feasible in AFM imaging considering the randomness of the interaction and its limited view. As it was manifested by the analysis of the SAXS data, the timing of the incorporation of NPs during the synthesizing process of LAFs did affect the morphological formation of the nanocomposites. In other words, the type-A process necessitates higher concentration of NPs before the deviations emerges when compared to the type-B process, of which the deviation appears even at the lowest concentration. In addition, these low-resolution models allow the best depiction of the distribution of compositional components, internal structural morphology and precise manners of NP aggregation in the various types of nanocomposites.

In summary, this combined SAXS and AFM approach provides a complementary perspective to studying the structure of Fe_3_O_4_ NPs-incorporated LAF nanocomposites and the effects of the NP sizes, volume concentrations and the timing of NP incorporation. The AFM measurements indicate that the addition of NPs of different sizes and concentrations to LAFs before and after the fibrillization process has a profound influence on the morphology and density of the attached clusters on the fibrillary surface. In general, the type-B process produces thicker nanocomposites with small NP sizes than the type-A process. Overall, the higher volume concentration promotes higher abundance of NP clusters alongside an increasing number of freely unattached NPs, which is especially apparent in the nanocomposites made by the type-A process. The SAXS results indicate that the formed NP clusters are highly dependent on the timing of NP infusion but relatively independent of the NP concentration. This study provides an alternative view into the formation of NP-fibril nanocomposites. It reveals a suitable range of NP concentrations to form such composites and underlines the importance of timing for the NP incorporation process. With such parameters, it appears possible to toggle the fibrillary aggregation at different stages of amyloidogenesis, before and after the fibrillization, corresponding to the entities of monomers and protofibrils. Such approaches could be potentially applicable to the treatment neurodegenerative diseases in future studies. Furthermore, such nanocomposites, which are formed under different parameters, may provide an alternative method for synthesizing novel electronic, optic or magnetic devices produced from bio-inorganic compounds [38,39,40].

## 4. Materials and Methods

### 4.1. Fe_3_O_4_ Nanoparticles

We used Fe_3_O_4_ magnetic NPs with the respective hydrodynamic sizes of 10 nm, 20 nm and 30 nm, with 5 mg/mL Fe concentrations and 0.02% NaN_3_, which were provided by and purchased from Ocean Nano Inc. (San Diego, CA, USA). These NPs were coated with an amphiphilic polymer coating composed of carboxylic acid atop an organic monomeric layer of oleic acid, and were dispersed in a borate buffer solution of 50 mM with pH of 8.5, and thus, were easily reactive to biomolecules. The sizes of the NPs were uniform within 10% variation and were highly stable within a pH range of 2 to 14, according to the manufacturer’s description.

### 4.2. Lysozyme Amyloid Fibrils

Lyophilized powder of hen egg white lysozyme (HEWL) with protein composition > 90% and molecular weight of 14.3 kDa (lyophilized power, lot number L6876, 50,000 units per mg protein) was purchased from Sigma-Aldrich Chemical Company (St Louis, MO, USA); other chemical reagents for preparation of the buffer solution (glycine, HCl and NaCl), from Sigma Aldrich (Saint-Louis, MO, USA), were all of analytic grade.

### 4.3. Sample Preparation

In brief, the synthetic procedure of LAF solution commenced with the dissolving of the HEWL powder in the buffer solution that consisted of 0.2 M glycine-HCl, with a pH of 2.2 and 80 mM NaCl, settling down at the concentration of 5 mg/mL. The buffer solution was conditioned at the acidic pH level of 2.2 for breaking down of the HEWL powder within a short period of time relative to the naturally occurring process. In the following step, two protocols were used to prepare the nanocomposite samples. To prepare type-A samples, the LAF powder solution was heated for 2 h at 65 °C while being constantly stirred at 250 rounds per minute (rpm) to form LAFs. The prepared LAF solution was doped with NPs of varying sizes and volume concentrations (10 μL, 30 μL, 300 μL) after the fibrilization process was finished (type-A process). Alternatively, the LAF powder solution was doped with NP of the same size and volume concentration, and was afterwards heated for 2 h at 65 °C while being constantly stirred at 250 rounds per minute (rpm) to form the fibrillary nanocomposites (type-B process).

### 4.4. Atomic Force Microscopy

Atomic Force Microscopy was chosen to study the surface morphology of the pure LAFs and their nanocomposites at nanoscale resolution. The AFM imaging presented in this work was performed using a 5500 AFM system (Agilent Technologies, Santa Clara, CA, USA) equipped with PicoView 1.14.3 control software (Agilent Technologies, Santa Clara, CA, USA) The images were acquired in tapping mode, using standard rectangular silicone tips with aluminum reflective coating on the back, with a nominal resonant frequency of 300 kHz and a nominal spring constant of 26 N/m (type OTESPA-R3, Bruker, Billerica, MA, USA). These tips had a visible apex with a very small radius (7–10 nm), ensuring a high lateral resolution to provide images of the fibrils. Samples for AFM observation were prepared by drop casting the LAF solutions on freshly cleaved mica sheets. The sample for AFM, which was actually a dried thin layer of fibrils on mica sheet, was placed on the AFM sample holder. Imaging of all samples was performed in an aerated environment with the humidity in range of 30–40% and at a room temperature of about 26 °C during the measurement. For each sample, a region-of-interest of 1024 × 1024 points was scanned. The captured images were processed using freely available software from Gwyddion (http://gwyddion.net, accessed on 1 January 2021).

### 4.5. Small Angle X-Ray Scattering

SAXS is sensitive to the structure and interactions of nanometer-sized particles in solution [31,32,33,41]. SAXS measurements were performed at the EMBL BioSAXS beamline P12 at PETRA III (DESY, Hamburg, Germany) [42] using a 150 µm × 250 µm (v × h, vertical × horizonal) X-ray beam of 10 keV energy (wave length λ = 0.124 nm). Sample solutions and the corresponding buffers, filled into quartz glass capillaries (outer diameter of 1.5 mm, wall thickness of 0.01 mm (Hilgenberg, Germany)), were placed into a temperature-controlled capillary holder, whose temperature was set to *T* = 20 °C. For each sample, a set of 20 frames with an exposure time of 50 ms were collected. Two-dimensional (2D) SAXS patterns were recorded using a PILATUS 6 M pixel detector placed at 3.0 m from the samples. With this experimental setting, a total *q*-range from *q* = 0.01–7.0 nm^−1^ could be accessed, with *q* denoting the scattering wave vector, which is defined as *q* = 4π/λsin(Θ), and 2Θ denoting the scattering angle.

### 4.6. SAXS Data Analysis

As all samples exhibited isotropic scattering patterns, the sets of frames for each sample were azimuthally averaged, normalized by the transmitted beam intensity, checked for radiation damage and averaged by the P12 beamline SASFLOW pipeline [43]. The background scattering signal from buffer solution was subtracted from these 1D-SAXS curves of the composite samples and the resulting SAXS profiles were further analysed using the ATSAS 3.0 software package [27]. Low resolution bead models from the SAXS data were obtained by performing ab initio modelling using the ATSAS programs, namely, DAMMIF, DAMMIX and DAMMIN (for details on the methods, see [29,36,37]). In short, these ab initio methods were able to determine low-resolution shapes from SAXS data by refining the particle shape using densely packed beads within a search volume without any prior information about the particles. Starting from a random configuration, simulated annealing was employed to fit the experimental data while ensuring the feasibility of the model (e.g., compactness and interconnectivity). Ten reconstructions, starting with random configurations, were performed and the resultant models were superimposed and analyzed. While the ab initio modelling approach is routinely used for biological SAXS, it has also, more recently, been applied to data from nanoparticles [44,45,46] and biological–nanoparticle hybrid systems [23].

## 5. Conclusions

We explored the nanometer-scale morphology of nanocomposites formed by Fe_3_O_4_ NPs and LAF by combining modern X-ray scattering with AFM. We determined the spatially distributed binding and morphology of NP clusters on the fibrillary surface and the influence of different parameters. The effects of the timing of NP infusion, and the volume concentration and size of NPs, on the internal structural morphology, the aggregating manner of NPs and NP clusters, and the 2D and 3D tomography of the nanocomposites, were studied.

The 2D surface topography measured by AFM confirmed that the interaction between Fe_3_O_4_ NPs and LAFs is dependent on the particle sizes and NP concentrations. Additionally, average ab initio models of the nanocomposites’ morphology were determined from the experimental SAXS data. Based on these beads-based nanocomposite models, the size, 3D spatial distribution, internal morphology, and manner of the affinity of the NPs on the fibrillary surface were quantified.

Our study demonstrated how the combination of modern solution SAXS approaches with real-space methods reveals structural information regarding biomaterials. With our combined approach, the adsorptive manner of the NP clusters on the fibrillary surface as a function of particle size, volume concentration, and infusion timing in the fibrillization process were determined. Modeling of the SAXS profiles revealed that the type-B process produces patchy, closely adhering adsorbates of NP clusters on the fibrillary surface, while type-A processes result in a less favorable interaction. For both processes and all conditions studied, LAFs were formed.

This work showed that the utilization of the effect of concentration and size on the nanocomposite structure can play an important role in the preparation of magnetically active bio-hybrid systems. Here, the understanding of NP–protein interactions is the first step towards realistic applications of these NP-mediated biological and biomedical effects and even towards strategic manipulations in vivo. While the use of iron oxide NPs in amyloid protein aggregation is still limited, because of their specific advantages, such as low toxicity and high magnetization, which allow for magnetic separation, Fe_3_O_4_ NPs are particularly interesting in terms of their potential clinical applications in relation to the accumulation of amyloid fibrils.

## Figures and Tables

**Figure 1 molecules-26-04864-f001:**

AFM images of pure LAFs at different resolution. Scale bars: (**a**) 4 µm, (**b**) 2 µm, (**c**) 500 nm.

**Figure 2 molecules-26-04864-f002:**
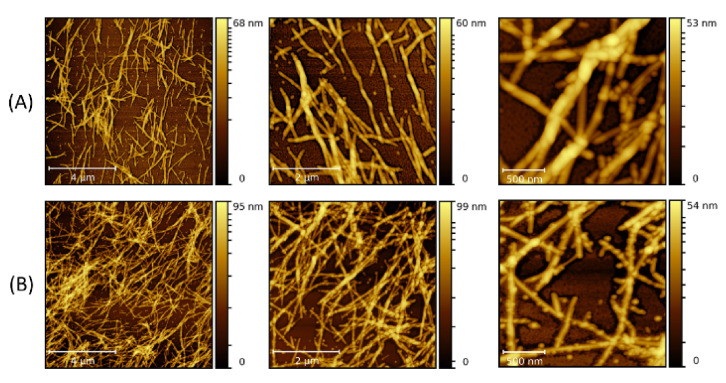
AFM images of the nanocomposites made by doping the LAFs with 10 nm NPs of (**A**) 10 μL and (**B**) 30 μL after the fibrillization process (type-A process). The scale bars are 4 µm, 2 µm and 500 nm from left- to right-hand panels.

**Figure 3 molecules-26-04864-f003:**
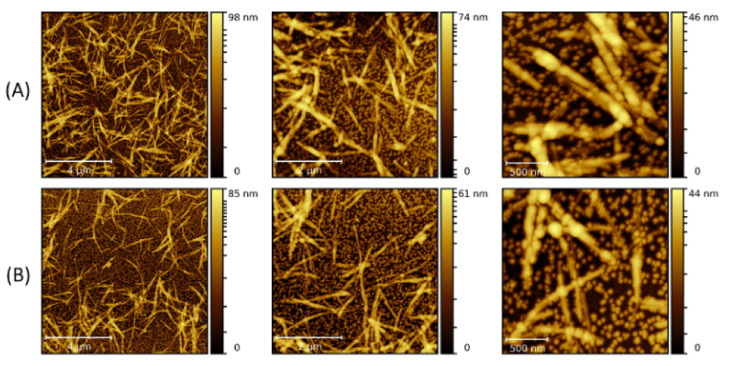
AFM images of the nanocomposites composed of LAFs and 10 nm NPs with volume concentrations of (**A**) 10 μL and (**B**) 30 μL by the doping of NPs before fibrillization process (type-B process). The scale bars are 4 µm, 2 µm and 500 nm from left- to right-hand panels.

**Figure 4 molecules-26-04864-f004:**
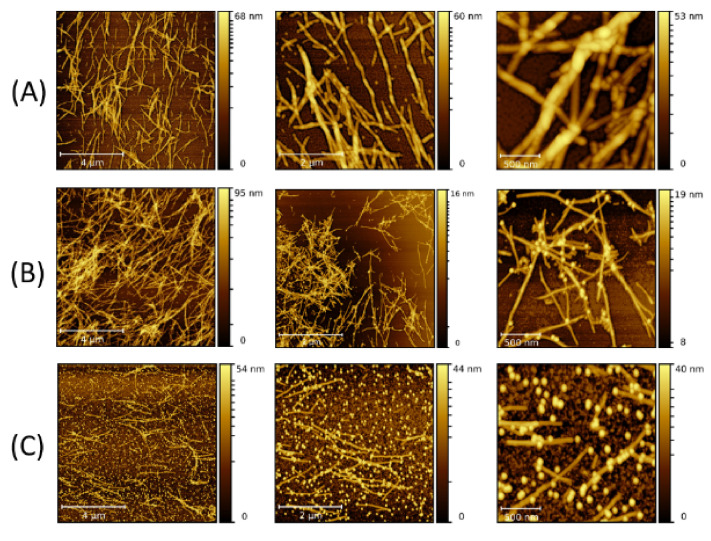
AFM images of the nanocomposites made by doping the LAFs with 20 nm NPs of (**A**) 10 μL, (**B**) 30 μL and (**C**) 300 μL after the fibrilization process (type-A process). The scale bars are 4 µm, 2 µm and 500 nm from left- to right-hand panels.

**Figure 5 molecules-26-04864-f005:**
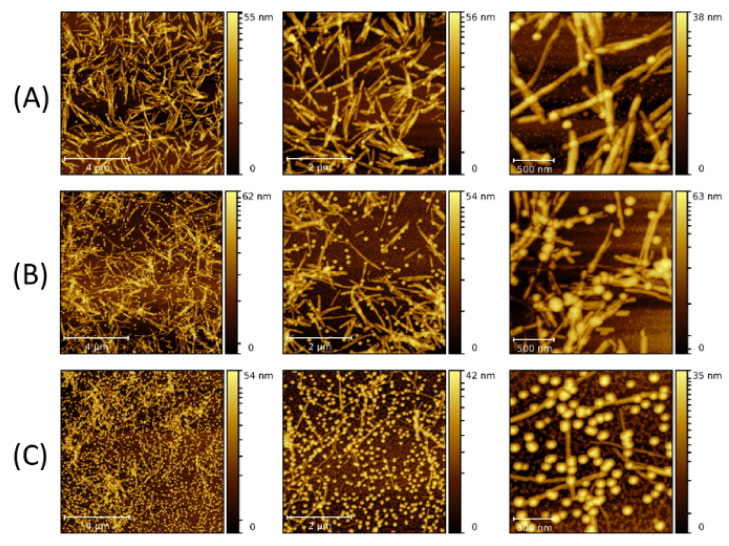
AFM images of the nanocomposites composed of LAFs and 20 nm NPs with volume concentrations of (**A**) 10 μL, (**B**) 30 μL and (**C**) 300 μL using type-B process. The scale bars are 4 µm, 2 µm and 500 nm from left- to right-hand panels.

**Figure 6 molecules-26-04864-f006:**
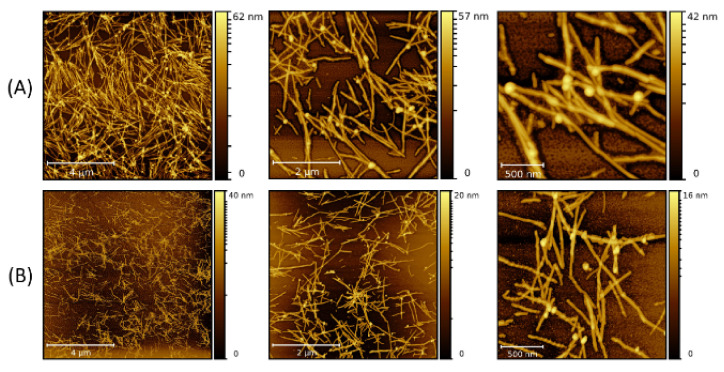
AFM images of the nanocomposites made by doping the LAFs with 30 nm NPs of (**A**) 10 μL and (**B**) 30 μL after the fibrillization process (type-A process). The scale bars are 4 µm, 2 µm and 500 nm from left- to right-hand panels.

**Figure 7 molecules-26-04864-f007:**
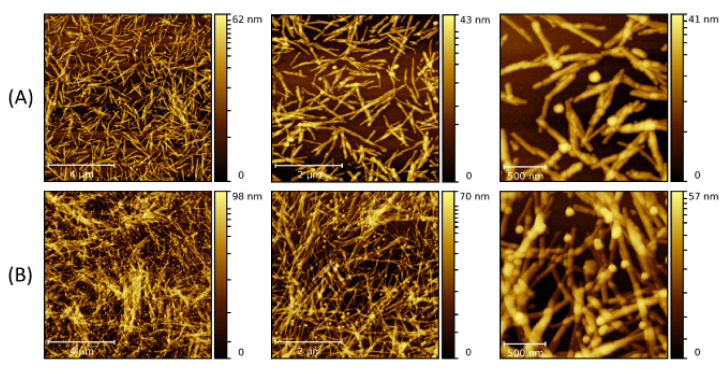
AFM images of the nanocomposites composed of LAFs and 30 nm NPs with volume concentrations of (**A**) 10 μL and (**B**) 30 μL using type-B process. The scale bars are 4 µm, 2 µm and 500 nm from left- to right-hand panels.

**Figure 8 molecules-26-04864-f008:**
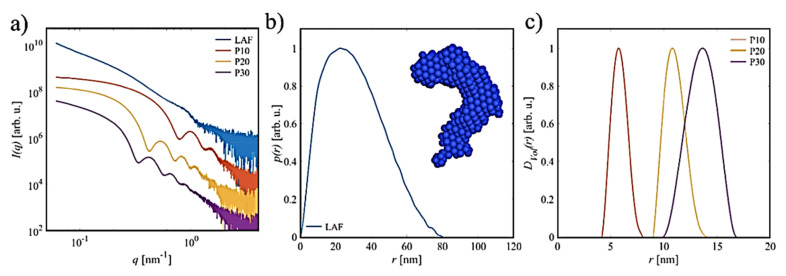
SAXS characterization. (**a**) The SAXS curves, *I*(*q*), of the solutions of LAFs and Fe_3_O_4_ NPs with nominal diameter sizes of 10 nm (P10), 20 nm (P20) and 30 nm (P30) (curves were shifted for clarity), (**b**) the pair–distance distribution function *p*(*r*) for LAFs alongside the corresponding ab initio shape model of the average solution structure of the fibril, (**c**) volume distribution function *D**_Vol_*(*r*) of the NPs.

**Figure 9 molecules-26-04864-f009:**
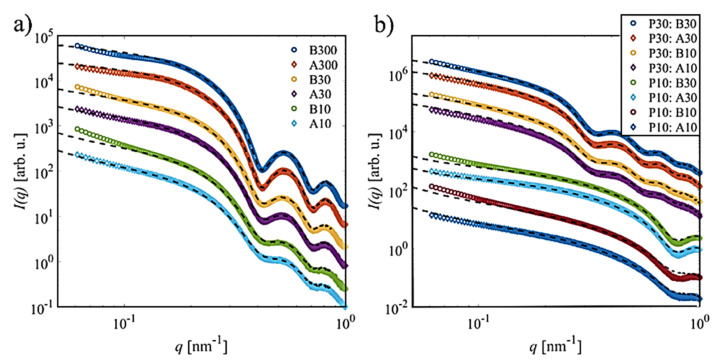
SAXS intensity profiles of the fibrillary nanocomposites. Scattering curves for the mixtures made of the LAFs with the infusion of (**a**) P20 at different volume concentrations, and of (**b**) P10 and P30 using type-B and type-A processes, are presented. Dashed lines are the fitting profiles of a hypothetical simple oligomeric mixture; the numbers next to the letters A and B indicate the volume of NPs added to the LAFs. All curves were shifted for clarity.

**Figure 10 molecules-26-04864-f010:**
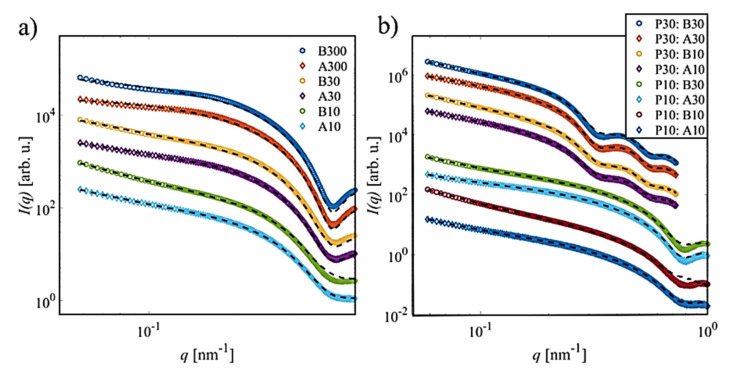
DAMMIX fits to SAXS profiles of the fibrillary nanocomposites. Mixture samples of (**a**) LAF and P20, and (**b**) LAF with either P10 or P30, respectively, are shown. Dashed lines represent DAMMIX fits; all curves were shifted for clarity. Due to the low-resolution of ab initio modelling, a different *q*-range was used for the DAMMIX analysis.

**Figure 11 molecules-26-04864-f011:**
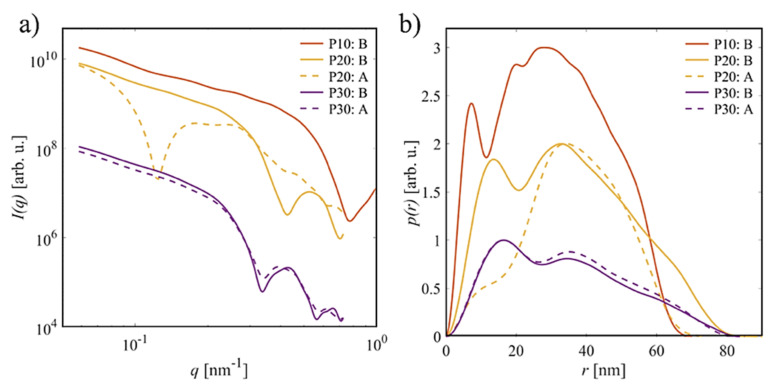
(**a**) Computed SAXS curves for each dataset of the intermediate clusters. (**b**) Corresponding *p*(*r*) function. All curves were scaled for clarity; for P10A, no intermediate is present.

**Figure 12 molecules-26-04864-f012:**
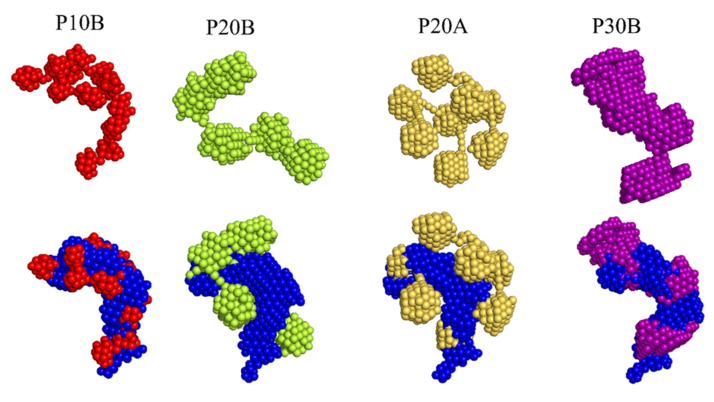
(**Top**) Ab initio shape models of the intermediate clusters determined from different series; (**bottom**) intermediate models aligned and overlaid with the model of LAFs. The model for P30A is nearly identical to P30B and is, therefore, omitted.

**Figure 13 molecules-26-04864-f013:**
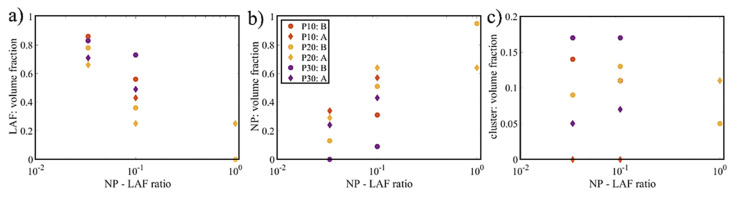
Volume fractions of (**a**) LAF, (**b**) NPs and (**c**) intermediate clusters in the nanocomposites as a function of NP/LAF ratio are presented.

**Table 1 molecules-26-04864-t001:** List of extracted structural parameters from SAXS for LAF and the NPs.

Sample	Size (nm)
LAF: maximum particle size *D**_max_*	80 ± 2
LAF: diameter *d*	23 ± 2
P10: 〈R〉	5.7 ± 0.1
P20: 〈R〉	10.7 ± 0.1
P30: 〈R〉	13.3 ± 0.4

## Data Availability

The data presented in this study are available within this article. Further inquiries may be directed to the authors.

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
