# Peer review of "Dependence of the Nanoscale Composite Morphology of Fe3O4 Nanoparticle-Infused Lysozyme Amyloid Fibrils on Timing of Infusion: A Combined SAXS and AFM Study"

_molecules, 2021, doi:10.3390/molecules26164864_

Round 1

Reviewer 1 Report

The manuscript titled Dependence of the nanoscale composite morphology of Fe3O4 nanoparticle-infused lysozyme amyloid fibrils on the timing of infusion: A combined SAXS and AFM study describes the application of SAXS and AFM imaging to evaluate the surface topography and internal structure of the nanocomposites formed by LAF and Fe3O4 NPs of different sizes as well as the role of mixing time-point of NPs during the fibrillization process. Although the idea of study seems to be very interesting, the purpose of research carried in this paper is not clear, and I cannot find the direct aim of the study. Authors present their findings to prove the applicability of AFM and SAXS in the interaction between lysozyme amyloid fibrils (LAFs) and Fe3O4 NPs before and after fibrillization, but a direct link between results and their biological relevance is missing.

  • The main goal of the work should be clarified, and fragment how the presented results could be used in the term of solving the biological problem should be added to the introduction
  • In the introduction section the statement: “Up-until-now numerous studies” (page 2, line 71) was supported by only 2 manuscripts that do not review. In my opinion, it is not enough
  • In the introduction section, I missed the conclusions from cited papers, e.g. the fragment “More recently in our previous work [22], the structural morphology of nanocomposites made from lysozyme amyloid fibrils (LAFs) and Fe3O4 NPs was studied. In particular, the distribution of NP aggregates on the surface of LAFs was revealed“ (page 2, line 68) was not commented what was the results and/or conclusions
  • Is it possible to quantify the area of mica covered by LAFs to numerically present the differences between samples?

In the section „Materials and Methods” the Authors described the steps of the preparation and measurement of samples. Nevertheless, the experimental section about AFM should be supplemented by the answers to additional questions listed below:

  • What kind of AFM tip do you use? What shape and size of AFM tip?
  • The force constant suggests a very hard tip, can the authors add an explanation as to why they chose a tip with such a force constant for research. Also, the method of spring constant calibration should be provided.
  • What is the resolution of AFM images (the size vs the number of points)?
  • How many samples were measured? How many AFM images were acquired from one sample? How many AFM images do you have in total?

The manuscript should also be edited, as there are many editorial issues e.g.: double spaces (page 12, line 375 or 385), unnecessary fragments (page 12, line 381). Also “ab initio” (page 4, line 14) should be in italics

In summary, I listed a lot of shortcomings of the submitted paper. Despite the interesting subject of the manuscript and the use of SAXS and AFM imaging to evaluate the surface topography and internal structure of the nanocomposites formed by LAF and Fe3O4 NPs of different sizes, the manuscript is not precisely outlined and described. In my opinion, in the manuscript, a solution to the scientific problem is missing. Due to the all above-highlighted aspects, the manuscript in the present form cannot be recommended for publication in Molecules, however, I would reconsider publication after major revision is done, and all of the remarks given herein are addressed by the authors.

Reviewer 2 Report

Mayor revision:

  1. Authors show many AFM images without deep explanation. The Fig.1 show three different images of the same sample with the same information. Why authors show these three images? Which different information we can extract from these only different size images?
  2. Please identify the average size of LAF in the first AFM images (diameter and longitude). Which is homogeneity of these fibers? They are agglomerate of the mica surface or not? Please use the AFM images for give readers adequate information.
  3. In the case of nanocomposites with Fe3O4, the NP are not only on the surface of the LAF but also on the mica surface, How you are sure that the NPs are anchored on the LAF surface. You need to proof it. In the present form you have surface covered with the NPs what do nit means that they are attached the LAF surface.
  4. What about distribution of NPs on the LAF surface from the Figure 2-4? Do you think that you need so many different AFM size images to proof the same?

Round 2

Reviewer 1 Report

I have no further comment. I recommend the manuscript for publication in the present form.

Reviewer 2 Report

Publish as is.